# Educational inequalities in mortality amenable to healthcare. A comparison of European healthcare systems

Håvard T. Rydland[1]*, Erlend L. Fjær[1], Terje A. Eikemo[1,2], Tim Huijts[3], Clare Bambra[4], Claus Wendt[5], Ivana Kulhánová[2], Pekka Martikainen[6], Chris Dibben[7], Ramunė Kalėdienė[8], Carme Borrell[9,10], Mall Leinsalu[11,12], Matthias Bopp[13], Johan P. Mackenbach[2]

1 Centre for Global Health Inequalities Research (CHAIN), Department of Sociology and Political Science, Norwegian University of Science and Technology (NTNU), Trondheim, Norway, 2 Department of Public Health, Erasmus MC, Rotterdam, The Netherlands, 3 Research Centre for Education and the Labour Market, Maastricht University, Maastricht, The Netherlands, 4 Population Health Sciences Institute, Newcastle University, Newcastle, United Kingdom, 5 Sociology of Health and Healthcare Systems, University of Siegen, Siegen, Germany, 6 Population Research Unit, University of Helsinki, Helsinki, Finland, 7 School of Geosciences, University of Edinburgh, Edinburgh, United Kingdom, 8 Lithuanian University of Health Sciences, Kaunas, Lithuania, 9 Agència de Salut de Pública de Barcelona, Barcelona, Spain, 10 CIBER of Epidemiology and Public Health, Madrid, Spain, 11 Stockholm Centre for Health and Social Change, Södertörn University, Huddinge, Sweden, 12 Department of Epidemiology and Biostatistics, National Institute for Health Development, Tallinn, Estonia, 13 Epidemiology, Biostatistics and Prevention Institute, University of Zürich, Zürich, Switzerland

* havard.rydland@ntnu.no

## Abstract

### Background

Educational inequalities in health and mortality in European countries have often been studied in the context of welfare regimes or political systems. We argue that the healthcare system is the national level feature most directly linkable to mortality amenable to healthcare. In this article, we ask to what extent the strength of educational differences in mortality amenable to healthcare vary among European countries and between European healthcare system types.

### Methods

This study uses data on mortality amenable to healthcare for 21 European populations, covering ages 35–79 and spanning from 1998 to 2006. ISCED education categories are used to calculate relative (RII) and absolute inequalities (SII) between the highest and lowest educated. The healthcare system typology is based on the latest available classification. Meta-analysis and ANOVA tests are used to see if and how they can explain between-country differences in inequalities and whether any healthcare system types have higher inequalities.

### Results

All countries and healthcare system types exhibited relative and absolute educational inequalities in mortality amenable to healthcare. The low-supply and low performance

**Data Availability Statement:** The authors confirm that, for approved reasons, some access restrictions apply to the data underlying the findings. Our mortality data have been retrieved

from national statistical offices in the study countries. The original data can only be retrieved from each country directly due to protection of privacy. We have presented the sources of mortality data in a (S1 Table) with contact information for each study country. We confirm that others will be able to access the data in the same way as we did. We also confirm that there were no special access privileges.

**Funding:** This article is part of the HiNEWS project —Health Inequalities in European Welfare States— funded by NORFACE (New Opportunities for Research Funding Agency Cooperation in Europe) Welfare State Futures programme (grant reference: 462-14-110). For more details on NORFACE, see http://www.norface.net/11. EF and TAE were funded by the Norwegian Research Council sponsored project ESS R7 Health Module: Equality in the Access to Health Care (project number 228990). HTR received funding from the strategic research area NTNU Health in 2016-2019 at NTNU, Norwegian University of Science and Technology.

**Competing interests:** The authors have declared that no competing interests exist.

mixed healthcare system type had the highest inequality point estimate for the male (RII = 3.57; SII = 414) and female (RII = 3.18; SII = 209) population, while the regulation-oriented public healthcare systems had the overall lowest (male RII = 1.78; male SII = 123; female RII = 1.86; female SII = 78.5). Due to data limitations, results were not robust enough to make substantial claims about typology differences.

## Conclusions

This article aims at discussing possible mechanisms connecting healthcare systems, social position, and health. Results indicate that factors located within the healthcare system are relevant for health inequalities, as inequalities in mortality amenable to medical care are present in all healthcare systems. Future research should aim at examining the role of specific characteristics of healthcare systems in more detail.

## Introduction

Over the last few decades, many studies have shown that socioeconomic factors (such as educational attainment, occupational class, and income) are the leading determinants of population health in European countries, and their influence appears to have increased substantially (cf. [1–3]). Healthcare systems have been characterized as one of the key dimensions of modern welfare states, since welfare states constitute "a complex set of institutionalized citizenship rights", shaping "the causes and consequences of health, illness and healing" [4]. Nevertheless, healthcare has been by and large absent from major welfare state theories [5–9]. In this article, we explore and discuss the associations between healthcare and social inequalities in health, on the empirical basis of mortality data from 21 European countries.

Educational level and health are related through numerous pathways, such as smaller risk of unemployment, higher income, good housing conditions, low financial hardship, lower levels of health damaging behavior, and feelings of mastery, control, and social support [10]. Educational attainment is also closely related to health literacy: the ability to use reading and numerical skills to understand health information provided by for instance physicians, nurses, and pharmacists [11]. Educational inequalities in health and mortality appear to vary across European countries, with the rank order of countries depending on the indicator of health and mortality that is used (cf. [12–15]). Education is a pragmatic measure of social position status which is reasonably comparable across contexts, and often used in cross-national studies where data on income or occupation is unavailable or considered too context-dependent–as is the case with this article [16]. Further, education is less sensitive to reverse causation–for adults, educational attainment does not change if one's health deteriorates. Educational distribution in the study countries is available in S2 Table.

A common approach to comparative studies of and social inequalities in health has been to focus on the role of welfare regime types (e.g., [17]) or political systems (e.g., [18,19]). Welfare regime typologies have contributed to highlighting and comparing some of the principles underpinning welfare states, the generosity of social transfers, and entitlements and social rights, which all may affect the social distribution of health [20]. The results from this regime approach to health inequalities have been described as "a patchy picture with contradictory findings" [21].

A common criticism against the welfare state regime approach has been related to its crudeness–it has been argued that there is a need to specify which welfare state characteristics are of importance for public health outcomes [22]. Moreover, reviews of the regime approach to health inequalities have concluded that the empirical evidence does not consistently support the association between welfare regime and health outcomes proposed by welfare regime theory [21,23]. Most notably: The Nordic countries belonging to the Social Democratic welfare regime, committed to universality and equality, have exhibited high life expectancies in combination with comparatively large health inequalities–often described as the Nordic public health puzzle or paradox [15,20].

In order to further advance research on macro-level explanations for cross-national differences in socioeconomic health inequality, more detailed accounts of the specific aspects of welfare regimes or political systems most prone to influence health are needed. Further, there is a need to link specific country-level mechanisms to specific health outcomes rather than general indicators of health or mortality.

In this study, we aim to provide a novel contribution by exploring the variation of educational inequalities in mortality amenable to healthcare among European countries and healthcare system types. We argue that the healthcare system is a feature of welfare states that is most directly relevant and linkable to health outcomes, compared to for instance GDP per capita or indicators of healthcare spending. We further argue that mortality amenable to healthcare is a health outcome with a clearer and stronger connection to state or healthcare intervention than other measures of health and mortality [24]. Amenable mortality can be defined as deaths which are preventable through medical intervention and which should not occur in the presence of timely and effective healthcare, including prevention, diagnosis, and treatment [25–27]. From this perspective, we aim to explore variation across 1) European countries and 2) European healthcare system types.

## Welfare and healthcare typologies

Several strategies to measure and classify healthcare systems have been proposed since the 1970s, often based on healthcare expenditure, healthcare financing, service provision, and access regulation and resulting in versions of three healthcare system ideal types closely connected to Esping-Andersens welfare state regimes: voluntary insurance, social health insurance, and national health service [7]. Reibling, Ariaans, and Wendt [28] used 13 country-level variables to construct a typology of healthcare systems across 29 high-income countries. Health expenditure per capita and the number of GPs per population indicated healthcare supply, the financial and human resources spent on health. The role of the state and the public/private mix in healthcare was indicated by the public share of health expenditure, the share out-of-pocket payments, and the remuneration of specialists as a measure of cost sharing. Access regulation was measured by indicators of healthcare coverage and choice restrictions. Expenditure on outpatient-care and their GP-to-specialist ratio indicated primary care orientation. Finally, healthcare performance was measured by indicators of tobacco and alcohol consumption and a quality sum index based on avoidable hospital admissions. Here, tobacco and alcohol consumption were used as proxies for the effectiveness of a healthcare system's preventive efforts, as adequate data on regulatory and monitoring activities was not available. Factor analyses of these indicators resulted in a five-fold typology of healthcare systems (countries included in our data in bold):

Type 1 –Supply- and choice-oriented public systems (Australia, **Austria**, **Belgium**, **Czech Republic**, **France**, Germany, Iceland, Ireland, Luxembourg, **Slovenia**): Primarily public funded social insurance systems. Characterized by medium to high levels of financial and

human resources, free choice, and access regulation only by cost sharing. Performance scores are mediocre with regards to both prevention and healthcare quality.

Type 2 –Performance- and primary-care-oriented public systems (**Finland**, Japan, New Zealand, **Norway**, Portugal, South Korea, **Sweden**): Public funded high-performing healthcare systems. The state has a strong role in regulating access and in the payment of medical specialists. Primary care has high priority.

Type 3 –Regulation-oriented public systems (Canada, **Denmark**, **Italy**, Netherlands, **Spain**, **United Kingdom**): Primarily public funded healthcare systems. Medium level of resources, low levels of out-of-pocket payments, and high level of access regulation and limitation of choice. Lower priority of primary care and lower performance than Type 2.

Type 4 –Low-supply and low performance mixed systems (**Estonia**, **Hungary**, **Poland**, Slovakia): Mostly public funded healthcare systems with low levels of financial and human resources, high levels of out-of-pocket spending, strong access regulations, and low performance on prevention and quality of care.

Type 5 –Supply- and performance-oriented private systems (**Switzerland**, United States): Healthcare systems with a strong role of private financing and out-of-pocket payments. Public resources are in the majority, with high supply and expenditures. Access is regulated by sharing regulations such as deductibles. This type shows high quality-of-care performance.

Since we wanted to utilize the full range of our data, and to avoid calculating with single-country clusters, we grouped Lithuania (which is not included in the data of Reibling et al. [28]) in Type 4, and Switzerland (which is the only Type 5 country in our data) in Type 1. This is done based on an assessment of key indicators used in the initial factor analysis. Subsequently, only four of the five healthcare systems types were included in our analysis. As results from research using welfare state regimes to compare health inequalities have been largely inconclusive, our contribution with this article is to use a validated and more specific health outcome–amenable mortality rather than self-reported health or limiting longstanding illness–and a typology more directly related to health–Reibling and colleagues' [28] healthcare system types.

## Expectations

Our study design is not suited for predicting inequality effects of specific health policies. However, we expect inequality rates to vary across countries and healthcare system types, and results from previous research allow us to formulate some modest expectations with regards to this variation. First, low education can be associated with poor health by being an indicator of material disadvantage. Financial strain due to e.g. unemployment or low income may matter more in a context with scarce healthcare resources and high out-of-pocket payments. Blom, Huijts, and Kraaykamp's [29] analyses of repeated cross-sectional survey data revealed that high total and state provision of healthcare, measured as total and governmental healthcare expenditure, was associated with smaller educational inequalities in self-rated health, while specific inequality-reducing health policies had a less substantial effect. This leads us to expect that low public funding, as found in the low supply and low performance mixed systems (Type 4), is associated with higher levels of inequalities.

Second, the impact of strong access regulation and choice restriction, as found in the performance- and primary-care-oriented public systems (Type 1) and the regulation-oriented public systems (type 3), appears less clear. On the one hand, regulations may enhance health

equality, ensuring equal access and preventing overconsumption of services. On the other hand, to maneuver a bureaucracy-governed healthcare system may (unintentionally) reward immaterial resources typically associated with high socioeconomic position, such as health literacy, social networks and the ability to "work the system" [30].

Third, people of low socioeconomic position have tended to be more intensive users of general practitioners, mainly due to a higher disease prevalence [31,32]. High priority of primary care, as found in the performance- and primary-care-oriented public systems (Type 2), could therefore also be associated with lower inequalities.

## Data and methods

### Data

The EURO-GBD-SE project collected and harmonized mortality data from the 21 European countries for which comparable data was available. This article utilizes all available data, covering time periods between 1998 and 2006, depending on country (see S1 Table). This data is to our knowledge the latest individual-level mortality dataset encompassing a majority of European countries. The datasets included four Nordic countries (Finland, Sweden, Norway, and Denmark), six Western European populations (England & Wales, Scotland, Belgium, France, Switzerland, and Austria), four Southern European populations (Barcelona, Basque Country and Madrid (Spain) and Turin (Italy)), four Central/Eastern European countries (Slovenia, Hungary, Czech Republic, and Poland) and two Baltic countries (Estonia and Lithuania). The data covered the entire national, regional (Madrid, the Basque Country) or urban (Barcelona and Turin) populations. The data from Spain and Italy only covers parts of the population, which prevents us from generalizing to the whole countries. These populations are therefore excluded when we estimated relative and absolute inequalities for the different healthcare system types but are displayed in tables and figures as a reference point.

Mortality data for Hungary, the Czech Republic, Poland and Estonia came from cross-sectional (CS) unlinked mortality studies. Data for Barcelona and Madrid was derived from a cross-sectional census linked studies. Data for other European countries has a longitudinal design. In the cross-sectional unlinked mortality studies, information on socioeconomic position was derived separately from death certificates and census records. In the longitudinal studies, mortality was linked to socioeconomic position determined during a census. An overview of the mortality data sources is displayed in S1 Table.

The Finnish dataset included only 80% of the Finns. The Swiss dataset excluded Non-Swiss nationals, the French dataset excluded those born outside mainland and the Dutch dataset excluded people from institutions. The 100% linkage between the population and death registries was achieved in most of the included populations. In countries where the default in linkage was lower than 5% no corrections were applied. In countries and areas such as Austria, Barcelona, the Basque Country, and Madrid, where a higher percentage of deaths that could not be matched with the mortality registry, we introduced a correction factor. In Austria, the correction factor was broken down by sex and 5-year age group. In Barcelona, the Basque Country and Madrid, there were no variations by age and sex for excluded deaths. The correction factor was therefore equal to 1.06 (1/0.946) for Barcelona and the Basque Country and 1.25 (1/0.8) for Madrid.

The causes of death amenable to healthcare were selected on basis of the publications by Stirbu et al. (2010) and the AMIEHS (2011) report from the European Union's Public Health Programme. In public health research, the terms "avoidable", "amenable", and "preventable" have been associated with some ambiguity, and often been used interchangeably [33]. Piers, Carson, Brown, and Ansari [34] have argued that avoidable mortality includes amenable and

preventable conditions, where deaths can be averted from the former, while the latter can be prevented from occurring altogether. Others have attempted to classify mortality according to the relevant level of healthcare intervention: primary, secondary, and tertiary avoidable mortality [35], and health policy and medical care indicators of avoidable mortality [36]. For example, Perez and colleagues' [37] analysis of avoidable mortality in Spain showed that figures on avoidable mortality could be affected by different processes such as healthcare interventions, prevention and promotion strategies, or by intersectoral policies. The authors argued that the concepts (and sub-concepts) of amenable and avoidable mortality have tended to blur the image of the prevalence and trends of specific causes of death. Nolte and McKee [33] have further questioned the underlying assumption of these classifications: that health outcomes can be attributed to specific elements of healthcare. For several conditions, there are discrepancies in the literature regarding the effect of public health and medical interventions, and thus also the nature of their preventability. Additionally, the classification of amenable mortality may to a certain extent suffer from systematic cross-national variation in diagnosis, death certification, and cause of death classification [27]. When assessing amenable mortality in the different healthcare system types, we will also contrast these estimates with inequalities in all-cause mortality.

Our classification leans on the precedence set by previous cross-national comparisons of amenable mortality (cf. [38–40]). One contested measure has been to classify ischemic heart disease and heart failure as non-amenable. It has been argued that the impact of medical treatment on these causes of death is unclear, while the association with lifestyle factors such as smoking, alcohol consumption and obesity is strong. Causes of death classified as amenable to healthcare are reported in Table 1. Other scholars have used different versions of the same data with similar classifications. Stirbu et al. [41] found educational inequalities in mortality amenable to medical care across all European countries, particularly pronounced in Central-/Eastern-, and Baltic European countries; Plug et al. [42] found that these inequalities were not

**Table 1. Causes of death amenable to medical care according with ICD10 codes.**

| Cause of death | ICD10 codes |
|---|---|
| HIV/ AIDS | B20-B24 |
| Tuberculosis | A15–A19, B90 |
| Other infectious and parasitic diseases | A00-B99 |
| Cancer of colon-rectum | C18–C21 |
| Cancer of cervix uteri | C53 |
| Cancer of testis | C62 |
| Hodgkins lymphoma | C81 |
| Leukemia | C91-C95 |
| Rheumatic heart disease | I00–I09 |
| Hypertension | I10–I15 |
| Other heart disease | I30-I52 |
| Cerebrovascular disease | I60–I69 |
| Pneumonia/ influenza | J10–J18 |
| Asthma | J45–J46 |
| Appendicitis, hernia, cholecystitis and lithiasis | K11.5, K35-K38, K40-K46, K80, K81, N20, |
| Peptic ulcer | K27 |
| Prostate hyperplasia | N40 |
| Maternal deaths, conditions originating in the perinatal period | O00-O99 |
| Congenital heart disease | Q20-Q28 |

associated with inequalities in healthcare use; Mackenbach et al. [15] compared mortality ame-
nable to behavior change, amenable to medical intervention, amenable to injury prevention,
and non-preventable mortality, finding the smallest inequalities in the latter category, and the
steepest gradient in the former; Mackenbach et al. [43] found that mortality declined faster
among the higher than among the lower educated and that educational inequalities in mortal-
ity decline were similar between causes of death amenable to behaviour change and medical
care.

We used educational attainment as a measure of socioeconomic position. This was catego-
rized according to the International Standard Classification of Education as low (no or pri-
mary education and lower secondary education, ISCED 0–2), middle (upper secondary
education, ISCED 3–4) and high (tertiary education, ISCED 5–6) education. In order to create
comparability across countries, we needed the same educational grouping in all countries.
These three groups were what national educational classifications allowed us to create, and this
division is also utilized in the studies cited above. Table 2 displays the amenable mortality rates
by educational level.

## Analyses

All analyses were conducted separately for women and men aged 35–79 years (age interval
depending on country) and age-standardized with the European Standard Population as refer-
ence [44]. Individuals whose educational attainment was unknown were omitted from the
analyses. The magnitude of relative educational inequalities in mortality amenable to health-
care across European countries and across healthcare systems was calculated by relative indices
of inequality (RII) by means of Poisson regression. The RII is a regression-based measure that
accounts for the distribution of the population by educational groups using rank of educa-
tional attainment as a dependent variable [45]. The educational rank was calculated over all
three educational groups defined above. The resulted RII represents the risk of death at the
lowest educational level as compared to the highest educational level in the population. Values
larger than 1 indicate a disadvantage for the low educated, values smaller than 1 a disadvantage
for the high educated. The magnitude of absolute educational inequalities was calculated by
Slope Index of Inequality (SII), a regression-based measure that takes into consideration the
entire distribution of education; its values indicates differences in predicted values between
low and high educated. Positive values indicate a disadvantage for the low educated, negative
values a disadvantage for the high educated.

To further test the applicability of the different typologies, meta-analyses and analysis of
variance (ANOVA) was performed on RII and SII estimates. Meta-analyses are common in
systematic reviews and aim to synthesize data from multiple studies [46]. In this article, pooled
estimates were calculated for each healthcare system type through meta-analysis techniques;
each country estimate was weighed with its inversed variance to calculate effect summary with
standard errors and confidence intervals. Since the inequality rates were estimated from differ-
ent populations, we calculated random effects models when heterogeneity was not too low.
When performing ANOVA analyses, we used F-tests to compare the RII and SII means of the
healthcare systems, and to determine whether between-group variance was larger than within-
group variance. Meta- and ANOVA analyses utilize tests of statistical significance, but with a
small country-level sample size, estimates are bound to be surrounded by uncertainty [47]. We
therefore avoid using these analyses as tests of whether differences between healthcare system
types are significant or non-significant. Fig 1 displays statistical uncertainty as 95% confidence
intervals, while S3–S5 Tables includes the p-values from the ANOVA analyses.

**Table 2. Mortality rates by educational level standardized to the European Standard Population.**

| Country | Gender | Mortality rates, ISCED 0–2 | Mortality rates, ISCED 3–4 | Mortality rates, ISCED 5–6 |
|---|---|---|---|---|
| Austria | Men | 274.4 | 210.1 | 148.4 |
| | Women | 159.8 | 114.9 | 90.2 |
| Belgium | Men | 238.0 | 198.2 | 153.9 |
| | Women | 158.5 | 121.1 | 94.8 |
| Czech Republic | Men | 478.2 | 265.7 | 163.8 |
| | Women | 261.7 | 182.9 | 106.1 |
| Denmark | Men | 284.3 | 232.3 | 183.7 |
| | Women | 190.7 | 150.1 | 118.9 |
| England/ Wales | Men | 219.0 | 144.3 | 122.5 |
| | Women | 159.2 | 106.8 | 110.8 |
| Estonia | Men | 689.0 | 530.2 | 317.3 |
| | Women | 403.5 | 279.2 | 172.8 |
| Finland | Men | 242.7 | 184.1 | 138.6 |
| | Women | 144.9 | 102.8 | 74.7 |
| France | Men | 310.7 | 223.4 | 141.1 |
| | Women | 136.8 | 90.2 | 55.7 |
| Hungary | Men | 644.1 | 351.5 | 247.8 |
| | Women | 345.8 | 188.6 | 182.4 |
| Italy (Turin) | Men | 200.8 | 170.6 | 136.7 |
| | Women | 120.2 | 111.0 | 95.4 |
| Lithuania | Men | 405.4 | 270.4 | 155.2 |
| | Women | 235.6 | 130.3 | 73.9 |
| Norway | Men | 246.5 | 181.1 | 136.1 |
| | Women | 163.3 | 120.8 | 85.3 |
| Poland | Men | 248.6 | 134.6 | 84.1 |
| | Women | 130.5 | 78.6 | 48.2 |
| Scotland | Men | 223.2 | 163.3 | 148.7 |
| | Women | 158.2 | 72.8 | 99.4 |
| Slovenia | Men | 421.2 | 278.2 | 178.3 |
| | Women | 202.4 | 133.8 | 104.3 |
| Spain (Barc.) | Men | 239.3 | 193.2 | 151.6 |
| | Women | 119.5 | 94.3 | 78.1 |
| Spain (Basque) | Men | 206.5 | 162.6 | 158.1 |
| | Women | 95.1 | 77.6 | 67.8 |
| Spain (Madrid) | Men | 231.8 | 206.9 | 183.2 |
| | Women | 122.4 | 111.0 | 78.3 |
| Sweden | Men | 184.2 | 146.6 | 113.4 |
| | Women | 125.8 | 95.4 | 69.8 |
| Switzerland | Men | 183.6 | 113.8 | 83.5 |
| | Women | 88.4 | 61.4 | 46.5 |

## Results

Relative and absolute inequality estimates are displayed in Table 3. In all countries, and subsequently in all healthcare system types, RII > 1 and SII > 0, meaning that mortality amenable to healthcare was higher for lower educated groups in all populations, both in relative and absolute measures. Among men, Poland (RII 4.67) and the Czech Republic (RII 4.60) showed higher relative inequalities, while Denmark (RII 1.81) and Sweden (RII 1.95) showed the

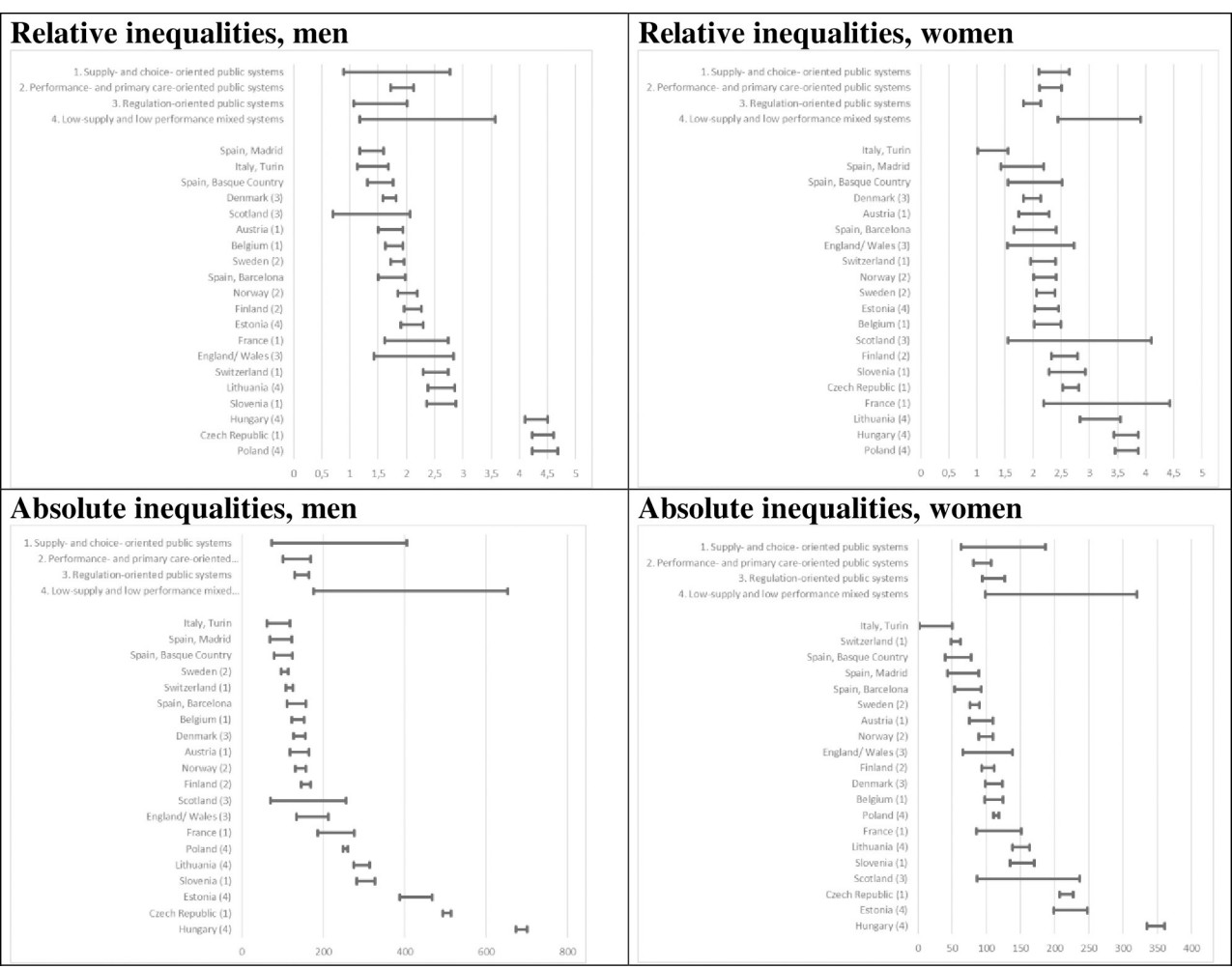

**Fig 1. RII and SII estimates (95% CIs).** Healthcare system types in parentheses.

lowest. The highest absolute inequalities were found in Hungary (683.3) and the Czech Republic (503.5), while the lowest inequalities were found in Sweden (SII 105.0) and Switzerland (SII 116.5). In the female population, Poland (3.66) and Hungary (3.65) showed the highest inequalities; Denmark (RII 2.0) and Austria (RII 2.0) has the lowest relative inequalities. Hungary (348.2) and Estonia 223.7) showed high absolute inequalities; Switzerland (55.2) and Sweden (82.9) had the lowest.

The healthcare system typology estimates were associated with much uncertainty and few clear-cut differences could be detected. A general pattern was that type 4, the low-supply and low performance mixed systems, had the highest point estimate in all analyses, while types 2 and 3, the performance- and primary care-oriented and the regulation-oriented public systems, showed the lowest absolute and relative inequality estimates respectively.

Results from ANOVA tests (S3–S5 Tables) were mixed; for most combinations of inequality measure and gender, except from relative inequalities among women, results indicated that variation between healthcare system types was not smaller than variation within types. These results imply that healthcare system similarities were not reflected in health inequality outcomes.

**Table 3. RII and SII estimates.** Standard errors in parentheses.

| | Men | | Women | |
|---|---|---|---|---|
| | **RII** | **SII** | **RII** | **SII** |
| Austria | 1.91 (0.11) | 141.4 (11.7) | 2.0 (0.14) | 92.2 (8.7) |
| Belgium | 1.93 (0.08) | 138.0 (7.8) | 2.25 (0.12) | 111.2 (6.8) |
| Czech Republic | 4.60 (0.10) | 503.5 (5.1) | 2.67 (0.07) | 217.4 (5.2) |
| Denmark | 1.81 (0.06) | 140.6 (7.2) | 2.0 (0.08) | 109.7 (6.5) |
| England/ Wales | 2.66 (0.36) | 171.6 (20.1) | 2.06 (0.3) | 100.6 (18.6) |
| Estonia | 2.28 (0.10) | 423.8 (20.5) | 2.23 (0.11) | 223.7 (12.7) |
| Finland | 2.26 (0.08) | 157.0 (6.0) | 2.55 (0.12) | 101.9 (4.6) |
| France | 2.62 (0.28) | 232.9 (22.9) | 3.12 (0.57) | 120.1 (16.7) |
| Hungary | 4.5 (0.1) | 686.3 (7.2) | 3.65 (0.11) | 348.2 (6.6) |
| Italy (Turin) | 1.64 (0.14) | 90.0 (14.5) | 1.25 (0.14) | 25.8 (12.4) |
| Lithuania | 2.84 (0.12) | 293.6 (10.1) | 3.18 (0.18) | 150.9 (6.3) |
| Norway | 2.18 (0.09) | 143.8 (6.7) | 2.2 (0.1) | 99.4 (5.4) |
| Poland | 4.67 (0.11) | 254.2 (3.0) | 3.66 (0.11) | 114.7 (2.0) |
| Scotland | 1.81 (0.35) | 162.6 (47.2) | 2.52 (0.65) | 164.6 (38.4) |
| Slovenia | 2.85(0.13) | 305.1 (11.7) | 2.58 (0.16) | 153.1 (9.0) |
| Spain (Barcelona) | 1.95 (0.12) | 134.4 (12.0) | 2.0 (0.19) | 71.7 (10) |
| Spain (Basque Country) | 1.73 (0.12) | 101.4 (11.4) | 1.98 (0.25) | 58.6 (9.6) |
| Spain (Madrid) | 1.57 (0.11) | 96.5 (13.8) | 1.77 (0.19) | 65.9 (11.8) |
| Sweden | 1.95 (0.06) | 105.0 (4.4) | 2.22 (0.08) | 82.9 (3.6) |
| Switzerland | 2.72 (0.11) | 116.5 (4.4) | 2.17 (0.11) | 55.2 (3.6) |
| Pooled estimate | 2.53 (0.22) | 220.1 (36.6) | 2.39 (0.14) | 123.0 (14.4) |
| 1. Supply- and choice-oriented public systems | 2.77 (0.48) | 239.6 (84.7) | 2.37 (0.14) | 124.9 (31.6) |
| 2. Performance- and primary care-oriented public systems | 2.12 (0.10) | 135.0 (17.2) | 2.31 (0.10) | 94.4 (6.6) |
| 3. Regulation-oriented public systems | 2.01 (0.24) | 146.8 (8.89) | 1.98 (0.08) | 110.6 (8.23) |
| 4. Low-supply and low performance mixed systems | 3.57 (0.61) | 414.5 (121.4) | 3.18 (0.37) | 209.3 (56.6) |

## Discussion

Few distinct conclusions can be drawn from our comparisons of European healthcare system types. As expected, Type 4 characterized by low supply in general showed the highest inequality rates, suggesting that high supply of healthcare services combined with focus on primary and preventive healthcare focus may moderate health inequalities. We outlined different mechanisms through which regulation of access and choice in a healthcare system could affect inequalities. The healthcare systems characterized by public financing and regulation of access had low point estimates of inequality. However, results were associated with uncertainty, demonstrated by the large confidence intervals. Type 4 scores low on both resources and the performance indicators, and it is thus difficult to distinguish any specific healthcare system characteristics affecting inequalities in amenable mortality. This inconclusiveness corresponds with the findings from Bergqvist, Yngwe, and Lundberg's [21] review, leading the authors to suggest that the regime approach "is not a fruitful way forward". In a sensitivity analysis (S6 Table), we calculated RII and SII estimates in total mortality for all countries and healthcare system types, finding similar patterns: The low-supply and -performance systems showed the largest relative and absolute inequalities, with indiscernible differences between the other types., results from ANOVA tests of all-cause mortality were, similar to those of amenable mortality, mixed. Greater variation was demonstrated between than among types only for relative inequalities among women and absolute inequalities among men. Analyses using all-cause

mortality accounts for competing causes; when using amenable mortality and excluding some causes of death, we risk removing data points where multiple morbidities have affected death. Results from these sensitivity analyses suggest similar population health patterns in the countries within each typology, but potentially through other mechanisms than similar healthcare systems.

Inequalities were demonstrated also in systems emphasizing high supply and state control of access and choice, i.e. being close to what one could call universal healthcare systems. A common explanation of health inequalities in these systems has been to emphasize social patterns in background risk factors, for example in smoking, since these systems exhibit large social inequalities in such risk factors [12,48,49]. However, we have defined mortality directly related to tobacco and alcohol (cancer of larynx, trachea, bronchus, and lung; chronic obstructive pulmonary disease; alcoholic psychosis, dependence, and abuse; alcoholic cardiomyopathy and cirrhosis of liver; and accidental poisoning by alcohol) as not amenable to healthcare, and thus excluded these causes of death from our analyses. This is not to say that smoking and drinking could not be indirectly related to other causes of death, for instance as cardiovascular-related mortality amenable to healthcare, but we have assumed them to only have a limited influence on the observed mortality inequalities, leaving the greatest explanatory power to factors located within the healthcare services.

Healthcare plays a key role in the social distribution of health, illness and death. Healthcare system arrangements may therefore function as mechanisms connecting social position to health outcomes. At the organizational level, a lack of access to good quality healthcare in lower socioeconomic groups could translate into larger educational inequalities in mortality. However, the evidence on this point is inconclusive, in particular for high-income countries with publicly financed healthcare systems [15,50]. A related, potentially inequality-producing, factor is unequal *use* of healthcare services by socioeconomic groups. Low socioeconomic position has been associated with more use of primary healthcare, while higher socioeconomic groups have reported significantly more specialist contact, even though they overall are in better health. These inequalities have been shown to vary across countries and welfare state regimes [31,32,51–53]. Some examples of suggested explanations are 1) that physicians could be more concerned about high-status patients; 2) that low-status patients are less able to "work the system" and pressure their physicians to prescribe more care; 3) that the interpretation of symptoms and perception of the need for healthcare, are closely associated with socioeconomic position; and 4) that patients with low education are more sensitive to a paternalistic doctor-patient relationship [30,54–56]. At the level concerning the specific treatment and the physician-patient relation, patients with low education and patients who in less affluent areas are more likely to receive shorter primary care consultations and to experience their physician as less empathic [57,58]. Similar to previous research, our results indicated that amenable mortality inequalities existed in all study countries and healthcare system types. The type characterized by low resources and access regulation showed signs of the overall largest inequalities, but some decoupling of the typologies is still needed. Further, our data did not allow us to determine whether these inequalities estimates stem from inequalities in access, in use, or in quality of healthcare services.

## Limitations

The approach of classifying countries into typologies or regimes has been subject to debate. As Wendt [6] has demonstrated, several typologies with different healthcare system types and varying country classifications have been proposed during the last few decades (e.g. [7,59–61]). Although typologies inherently capture a broad range of interrelated dimensions, they also

always depend on the extent to which dimensions are emphasized or de-emphasized in the operationalization. Apparently similar programs and policies may be differently organized, and indicators upon which a typology is based, for instance choice restrictions and funding, may be confounded. However, the healthcare system typologies first developed by Wendt [6] and later followed up by Reibling et al. [28] is to our knowledge the most comprehensive typology to our knowledge, aiming to intercept all important aspects of a healthcare system.

To adapt the Reibling et al. [28] typology to our available data material, we classified Lithuania and Switzerland as respectively *Low-supply and low performance mixed systems* and *Supply- and choice-oriented public systems*. Classification was done by key indicators from the initial factor analyses of Reibling et al. [28]. Additional meta-analyses and ANOVA tests showed that including these countries in their respective clusters affected meta-analysis estimates, but the overall differences between the estimates remained similar, while results from ANOVA tests excluding Switzerland and Lithuania indicated that the within-type variation was not lower than the between-type variation, similar to the analyses of amenable mortality.

Some compatibility issues occurred between the country-level healthcare system typology and the individual-level cause-specific mortality data. The Reibling et al. [28] typology is based on data from 2011 to 2014, while the mortality data covers the period 1998 to 2006 (depending on country, see S1 Table). Though the 2019 healthcare system types have similarities with earlier typologies (cf. [6,62]), this partial incompatibility weakens the link between our two data levels. Most all analyses combining data from the individual and country level face similar constraints; the influence of country-level variables on mortality is hard to narrow down in general, as numerous policies affect one's health over the life course. In our discussion, we have met this limitation by using the typologies to describe variations rather than assigning direct effects to specific policies.

The 20% of Finns excluded from the data was a random sample and results should not be affected. Related is the exclusion of non-Swiss nationals from the Swiss data. The impact of this potential bias is unclear; our analyses may over- or underestimate the magnitude of inequalities in mortality in Switzerland as a whole, depending on inequalities in mortality in the excluded population compared to Swiss nationals. As aforementioned, meta-analyses and ANOVA with and without Switzerland returned similar results, but this exclusion nevertheless limits our conclusions. Non-linkage represents another limitation; applying the correction factor provides a more accurate result but will not remove a systematic non-linkage bias–we do not know the composition of the non-linked populations. Lastly, the "No education" and "Missing education data" categories may be heterogenous; Flanagan and McCartney [63] have demonstrated how differentiation across categories and missing data on educational attainment has varied between censuses in England and Wales from 1971 to 2001. The ISCED categories provides comparability across countries, but national differences in questioning, coding, and organization of the education system are still unaccounted for.

The applied definition of amenable mortality and the indicators used to construct a typology may also be conflicting. An apparent example is that consumption data on alcohol and tobacco are used to measure for healthcare prevention performance, while mortality directly related to lifestyle traits was excluded from the analyses. Variation in countries' performance in preventing smoking and alcohol use may thus not be reflected in the mortality numbers. On the other hand, Reibling et al. [28] included these indicators as proxies; they are meant to indicate general preventive care performance. Further, only mortality *directly* attributed to smoking and alcohol use was excluded; we included causes of death *indirectly* associated with lifestyle, which again could be related to the performance of a country's preventive services.

The concepts of amenable mortality and healthcare system types offers both the advantages and disadvantages associated with combining several dimensions in one encompassing

classification. Originally, amenable mortality was intended to be useful in terms of policy intervention, with an aim to distinguish those forms of mortality that a more effective organization of the healthcare system could deal with. However, such classifications may also hide variation between the different causes of death–within and across countries. Though amenable mortality was originally proposed as an indicator of healthcare quality, Nolte and McKee [33] have suggested–on the basis of the ambiguous operationalisations and evidence–that it rather should be treated as a starting point for further research and an indicator of concern. Although our analysis may suffer from crude divisions of mortality, we argue that these were necessary steps for the cause of overview and comparison, and as a point of departure for discussing how healthcare systems may produce health inequalities. We urge future research to derive more specific policy recommendations based on empirical analyses focusing on specific aspects of healthcare systems and detailed forms of amenable mortality. This will require the availability of rich data at the individual level as well as the national level for a large number of countries to improve statistical power.

## Conclusions

Many of the pathways connecting social position to health can potentially be found within the healthcare system. This article has combined a novel healthcare system typology with comprehensive individual-level mortality data. Our results demonstrated educational inequalities in mortality amenable to healthcare across 21 European populations. Meta-analyses suggested that higher inequalities were found in healthcare systems characterized by low healthcare supply, strong access regulation, and low scores on selected performance indicators.

All four healthcare system types exhibited inequalities in mortality amenable to medical care, and healthcare systems characterized by universality and high levels of provision did not show smaller inequalities. This paradox has previously been explained by pointing to inequalities in lifestyle traits, but our analyses indicated that inequalities are apparent in these systems also when mortality directly attributable to alcohol and tobacco is excluded, suggesting that organizational features of these healthcare systems also could be determinants of health inequalities, but the typology utilized may be a too crude measure. One purpose of our analyses was to provide an overview and discuss how healthcare systems may affect health. We further recommend future research on amenable mortality and morbidity to examine specific health policies and their impact on specific amenable health outcomes.

## Supporting information

**S1 Table. Data sources.**
(DOCX)

**S2 Table. Educational distribution.**
(DOCX)

**S3 Table. Analysis of variance, RII and SII estimates of healthcare system types (amenable mortality).**
(DOCX)

**S4 Table. Analysis of variance, RII and SII estimates of healthcare system types–excluding Switzerland and Lithuania (amenable mortality).**
(DOCX)

**S5 Table. Analysis of variance, RII and SII estimates of healthcare system types (all-cause mortality).**
(DOCX)

**S6 Table. RII and SII estimates in total (all-cause) mortality.**
(DOCX)

## Author Contributions

**Conceptualization:** Håvard T. Rydland, Tim Huijts, Clare Bambra, Claus Wendt, Ivana Kulhánová, Pekka Martikainen, Chris Dibben, Ramunė Kalėdienė, Carme Borrell, Mall Leinsalu, Matthias Bopp, Johan P. Mackenbach.

**Data curation:** Håvard T. Rydland.

**Formal analysis:** Håvard T. Rydland.

**Methodology:** Håvard T. Rydland.

**Supervision:** Håvard T. Rydland, Terje A. Eikemo, Tim Huijts, Clare Bambra, Claus Wendt, Ivana Kulhánová, Pekka Martikainen, Chris Dibben, Ramunė Kalėdienė, Carme Borrell, Mall Leinsalu, Matthias Bopp, Johan P. Mackenbach.

**Writing – original draft:** Håvard T. Rydland, Erlend L. Fjær.

**Writing – review & editing:** Håvard T. Rydland, Erlend L. Fjær.

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
