## [Decision Letter · Decision Letter 0]

3 Mar 2020

PONE-D-19-34940

Educational Inequalities in Mortality Amenable to Healthcare. A Comparison of European Healthcare Systems

PLOS ONE

Dear Mr. Rydland,

Thank you for submitting your manuscript to PLOS ONE. After careful consideration, we feel that it has merit but does not fully meet PLOS ONE’s publication criteria as it currently stands. Therefore, we invite you to submit a revised version of the manuscript that addresses the points raised during the review process.

We would appreciate receiving your revised manuscript by Apr 17 2020 11:59PM. To enhance the reproducibility of your results, we recommend that if applicable you deposit your laboratory protocols in protocols.io, where a protocol can be assigned its own identifier (DOI) such that it can be cited independently in the future. For instructions see: http://journals.plos.org/plosone/s/submission-guidelines#loc-laboratory-protocols

A **rebuttal letter** that responds to **EACH** point raised by the academic editor and reviewer(s). This letter should be uploaded as separate file and labeled 'Response to Reviewers'.A **marked-up copy** of your manuscript that highlights changes made to the original version. This file should be uploaded as separate file and labeled 'Revised Manuscript with Track Changes'.An **unmarked version** of your revised paper without tracked changes. This file should be uploaded as separate file and labeled 'Manuscript'.

We look forward to receiving your revised manuscript.

Kind regards,

Brecht Devleesschauwer

Academic Editor

PLOS ONE

Additional Editor Comments (if provided):

In your revision note, please include EACH comment of the reviewers, provide your reply, and when relevant, include the modified/new text (or motivate why you decided not to modify the text). Note that failure to do so may result in a rejection of the manuscript.

Journal Requirements :

2) Please ensure that you include a title page within your main document. We do appreciate that you have a title page document uploaded as a separate file (cover letter), however, as per our author guidelines (http://journals.plos.org/plosone/s/submission-guidelines#loc-title-page) we do require this to be part of the manuscript file itself and not uploaded separately.

3) Please include a copy of Table 4 which you refer to in your text on page 8.

4) One of the noted authors is a group or consortium NORFACE HiNEWS project and the EURO-GBD-SE Consortium. In addition to naming the author group, please list the individual authors and affiliations within these groups in the acknowledgments section of your manuscript. Please also indicate clearly a lead author for these groups along with a contact email address.

5) We note that you have indicated that data from this study are available upon request. PLOS only allows data to be available upon request if there are legal or ethical restrictions on sharing data publicly. For information on unacceptable data access restrictions, please see http://journals.plos.org/plosone/s/data-availability#loc-unacceptable-data-access-restrictions.

Reviewers' comments:

Reviewer's Responses to Questions

**Comments to the Author**

1. Is the manuscript technically sound, and do the data support the conclusions?

Reviewer #1: Partly

Reviewer #2: Partly

2. Has the statistical analysis been performed appropriately and rigorously? 

Reviewer #1: Yes

Reviewer #2: I Don't Know

3. Have the authors made all data underlying the findings in their manuscript fully available?

Reviewer #1: No

Reviewer #2: Yes

4. Is the manuscript presented in an intelligible fashion and written in standard English?

Reviewer #1: Yes

Reviewer #2: Yes

5. Review Comments to the Author

Reviewer #1: Thank you for the opportunity to review this manuscript. This is the latest in a series of articles which uses mortality data stratified by educational attainment for a selection of European countries to examine the relationship between relevant exposures and inequalities in mid-life mortality.

Detailed comments are provided below. Overall, it is a very interesting paper and addresses an important question. There are numerous limitations with the exposure and outcome measures, and with the potential for confounding. These all limit the inferences that can be made.

I am sorry there are so many comments. It is worthwhile research!

Abstract:

1. The use of 'tertiles' is misleading as the three educational groups are not equally sized. Instead the authors should just say three groups. It would be helpful (in the body of the main manuscript) if the authors could confirm that they have taken the difference in size of the educational groups within each country into account in calculating the RII and SII values.

2. The authors should mention the time period for which the mortality data pertain in the methods.

3. It would be useful to provide some numerical results in the abstract, perhaps summarising the meta-analytical results.

4. I will make further comments on the use of significance testing elsewhere, but I do not think the use of significant in the abstract is appropriate.

Introduction

1. In setting up the research question the authors use educational attainment as a pragmatic means of ranking the population to assess health inequalities. This should be made explicit that there is no attempt to use any theories of socioeconomic position or class to understand how educational attainment per se is related to the outcomes. This is okay for the purpose of this paper, but it would be helpful for this to be explicit.

2. The authors have misrepresented Julian Tudor Hart's seminal work on the inverse care law in the second paragraph. The full definition of this is, "The availability of good medical care tends to vary inversely with the need for it in the population served. This inverse care law operates more completely where medical care is most exposed to market forces, and less so where such exposure is reduced. The market distribution of medical care is a primitive and historically outdated social form, and any return to it would further exaggerate the maldistribution of medical resources." (https://www.thelancet.com/journals/lancet/article/PIIS0140-6736(71)92410-X/fulltext). This is important because the inverse relationship is explicitly linked to the degree of marketisation of the healthcare system - something that is only superficially addressed in the composite index used for the analysis in this paper. I did not follow the description and application provided here and would urge the authors to reconsider and redraft this section.

3. I understand that the focus of the work here is in examining inequalities rather than mean outcomes. However, it would be helpful to acknowledge in paragraph three that the Nordic countries have comparably high life expectancy despite reasonably high inequality.

4. (Relevant to the methods and discussion section too.) The definitions of amenable and avoidable mortality have been developed in order to distinguish the role and contribution of healthcare interventions. There is some discussion in the paper about these and the authors have taken the decision to change the accepted definitions in order to reduce the role of some causes (e.g. those closely related to substance use) as they view these are less amenable to healthcare intervention. In doing so the authors are implicitly acknowledging that none of the definitions of amenable or avoidable mortality are very accurate in that almost all causes of death are both socially/economically caused, and, to some degree, amenable to some healthcare intervention. It would be helpful to have a fuller discussion of what the use of the selected mortality codes do and do not show, and the limitations of these. It would be particularly helpful to discuss how this fits with the concepts of primary, secondary and tertiary prevention.

5. The discussion on the different means of classifying healthcare systems is very informative and interesting. It does however highlight the limitations of this way of classifying (similar to the limitations of the classifications of welfare state types). I am not an expert in the variety of healthcare systems across Europe but I did think that some of the groupings were very surprising. The link between restricting choice (which is an important means of preventing ineffective spending on low impact pharmaceuticals and specialist input) and funding is confounded - as the former limits the latter all other things being equal. The very small number of countries for which data were used in each category very much limits the extent to which it can be said that any differences in outcomes are due to the systems or other unmeasured confounding factors. Bringing together categories seems to further blunt any nuance within the typology and is another limitation that needs acknowledged.

Hypotheses

1. Please edit the sentence with "...supply-healthcare systems Even through these systems...".

2. I wasn't clear about the logic of the hypothesis that "high public involvement in the healthcare system and of high supply, free access and choice does not result in high educational inequality in amenable mortality". This runs counter to Tudor Hart's work about marketisation of healthcare. This may depend what is meant by 'public involvement', as the meaning of this is not clear to me.

Methods

1. Please justify why the selection of countries was made. Were these the only countries for which data was available perhaps?

2. The data used in the analysis is now quite old. This needs to be acknowledged and justified.

3. The exclusion of 20% of Finns and non-Swiss nationals needs to be discussed further. To what degree do these create systematic biases?

4. The correction factor used will not correct for systematic biases in non-linkage. This needs to be discussed as a limitation.

5. I wasn't convinced by the claims that ischaemic heart disease and heart failure are not amenable to healthcare intervention. Clearly it has both social and economic causes as well as being amenable to treatment. This again highlights the limitations of the amenable mortality measure and the attempt to dissociate the healthcare effects from the social and economic effects.

6. Could the authors justify why a pre-analysis protocol was not produced and published online?

7. I was unclear as to why the ISCED categories where grouped into three groups rather than used across all available data to service the regression modelling. Can this be justified?

8. Omitting those without educational attainment recorded is problematic. For example, in some years in the UK this has been as high as 85% of the population and systematically different from the population mean (see https://www.ncbi.nlm.nih.gov/pubmed/25862252).

9. It is described that mortality was "controlled for 5-year age groups". Do the authors mean that the data were age-standardised? If so, to what standard population?

10. The approach to the ANOVA analysis is well-described and appropriate.

11. I found it somewhat implausible that cerebrovascular disease was classified as amenable but IHD was not. I worked as a GP at this time and if anything the opposite was true.

Results

11. The authors are over-reliant on statistical significance to make inferences (see https://www.bmj.com/content/322/7280/226.1). There is substantial confusion in the reporting about whether they are simply talking about the degree to which there could be random variation; small sample sizes and underpowering of the analysis; and the importance/policy relevance of the analysis. These are all conflated and strong conclusions are made about there being no differences in some cases when in fact this is simply likely to be due to small samples.

12. It would be best if the ANOVA results were simply put into a webappendix so that it can be readily available in the future and fully transparent. People move on and analyses get lost otherwise!

13. It is important that the mortality rates for each ISCED group, and the proportion of the population in each ISCED group, and the mean for the total population, are provided in a table.

14. The sensitivity analyses should be shown in a webappendix.

Discussion

1. In the first sentence the authors discuss equal access to healthcare. Do they mean equal or equitable? Being clearer about their definition of health inequality would also be helpful in this regard.

2. The use of Barcelona, Basque Country, Madrid and Turin is a major limitation and the authors should not make any generalisation to the countries overall in the use of these data. The points made about urban and rural inequalities are not relevant here. It is akin to generalising to the whole of the UK from London or Glasgow inequalities, either of which would be completely misleading. Within Spain the poorest regions are not included; the same is true for Italy.

3. At the end of the first paragraph ("In line with more agency-based approaches...") the formulation seems to be that regulation is about preventing the rich using their resources to access healthcare. This seems a little inconsistent with the countries in the typology where there is a private healthcare sector which regulation does not limit access to. I'd urge the authors to redraft this section.

4. The second paragraph discusses some aspects of the inverse care law. There is a large primary care literature on this (see Graham Watt, Stewart Mercer, Mhairi Mackenzie and others for example) which could usefully be integrated into this section.

5. I think the robustness of the results of the paper are overplayed and the limitations under-recognised in the discussion. There are numerous issues with the measures of the exposure and outcome, as well as limited data availability and a high risk of confounding. Taken together it is quite difficult to be sure that the relationship described is robust.

6. A limitation not recognised in the use of a selection of mortality codes is competing causes. By removing IHD etc., people with multiple morbidity in middle age will be removed from the analysis and underestimate the mortality rates that would have occurred had the IHD not intervened. Comparing the results with all-cause mortality inequality would allow this to be discussed in more detail.

Table 1

1. I was not clear what the numbers in the period column mean. Possibly months and years?

Figure 1

1. It is not clear what the vertical dotted lines represent.

Reviewer #2: Summary

Using data on mortality amenable to health care for 21 European populations, this manuscript presents estimates of relative and absolute educational inequalities and investigates them through the lens of European health systems. The study advances on previous research which focused primarily on welfare typologies and/or aggregates of political systems. The authors find considerable variation across the clusters of health systems. Further analyses suggest that health care budgets, rather than access regulation or choice control, may explain some of these differences.

There is a lot to like about this manuscript since it examines the role of health systems in one closely linked outcome, notably, educational inequalities in mortality amenable to healthcare. It draws on a recent typology and discusses a range of causal pathways. In short, the manuscript promises to be a valuable contribution to a growing literature on health inequalities. At the same time, I believe several important issues—pertaining to the typology, data, and design—need to be addressed before publication.

---

Major issues

The motivation of the study, focusing on specific elements of welfare regimes (in this case, health systems) rather than broader categories, is first discussed in the fourth paragraph. I think it would be helpful to lead with health systems, and then conceptualize their role in the broader discussion on health inequalities.

While the authors do a good job describing the typology of health systems, I believe one key aspect is missing: temporal information on the underlying variables. That is, the typology is based on measures that vary considerably over time, e.g., health expenditure per capita, number of GPs, public share of health expenditure, tobacco and alcohol consumption, etc. However, it is unclear which years were used to construct the original typology which raises at least two concerns: First, the typology might be based on recent data, which means that educational inequalities based on mortality data from ca. 2000-2005 (depending on the country, according to Table 1, p. 22) are related to characteristics of ‘current’ health systems. Second, if the typology is identified in years around the Millennium, i.e., similar to the health inequality measures, the implications for current health governance are different. In any case, I find the reader is unable to evaluate the analysis and conclusion due to this uncertainty.

Data issues are another major concern for me. First, some of the data cover regional or urban populations. However, the hypotheses and typology of health systems are based on national characteristics (e.g., to what extent is the population of Madrid and/or its health system measures representative of Spain?, do these drive results?). Second, ischemic heart disease and heart failure were excluded due to their ‘strong association with life style factors such as smoking, alcohol consumption and obesity’ (p. 7). Yet for the typology of health systems, ‘healthcare performance was measured by indicators of tobacco and alcohol consumption …’ (p. 4). Thus, the performance measure does consider tobacco and alcohol, the mortality data does not. I think this inconsistency needs to be addressed.

The authors argue for the benefits of using health system typology as explanatory variable, while acknowledging several limitations (pp. 13-14). Particularly in view of the finding that health spending is important, I remain to be convinced why, under these circumstances, the typology is useful as an analytical category. In the concluding section, the authors ‘recommend future research on amenable mortality and morbidity to examine specific health policies’ (p. 16). Why does this study not already cover at least one of these aspects?

---

Minor issues

In the opening paragraph (p. 2), the authors mention numerous pathways through which education affects health outcomes and behavior. I would encourage them to use these insights and studies, and discuss the hypotheses (pp. 5-6) with a stronger focus on educational inequalities.

In Figure 1, it would be helpful for the reader if it was indicated which clusters the countries belong to. A note should also explain how to interpret the estimates.

6. PLOS authors have the option to publish the peer review history of their article (what does this mean?). If published, this will include your full peer review and any attached files.

Reviewer #1: Yes: Gerry McCartney

Reviewer #2: No

---

## [Author Response · Author response to Decision Letter 0]

28 Apr 2020

See uploaded manuscript for specific reviewer comments.

---

## [Decision Letter · Decision Letter 1]

20 May 2020

Educational Inequalities in Mortality Amenable to Healthcare. A Comparison of European Healthcare Systems

PONE-D-19-34940R1

Dear Dr. Rydland,

We are pleased to inform you that your manuscript has been judged scientifically suitable for publication and will be formally accepted for publication once it complies with all outstanding technical requirements.

With kind regards,

Brecht Devleesschauwer

Academic Editor

PLOS ONE

Additional Editor Comments (optional):

Reviewers' comments:

Reviewer's Responses to Questions

**Comments to the Author**

1. If the authors have adequately addressed your comments raised in a previous round of review and you feel that this manuscript is now acceptable for publication, you may indicate that here to bypass the “Comments to the Author” section, enter your conflict of interest statement in the “Confidential to Editor” section, and submit your "Accept" recommendation.

Reviewer #1: (No Response)

2. Is the manuscript technically sound, and do the data support the conclusions?

Reviewer #1: Partly

3. Has the statistical analysis been performed appropriately and rigorously? 

Reviewer #1: Yes

4. Have the authors made all data underlying the findings in their manuscript fully available?

Reviewer #1: No

5. Is the manuscript presented in an intelligible fashion and written in standard English?

Reviewer #1: Yes

6. Review Comments to the Author

Reviewer #1: Dear Authors

Thank you for the opportunity to reread your manuscript and comment again. It would have been very helpful if in your responses to the reviewers comments you had provided the quotes of the changed sections rather than just making general statements. Perhaps this is something you could do in the future to help reviewers.

I think you have generally understood the comments I made previously (I am reviewer 1), with the exception of the pre-analysis protocol (comment labelled Methods 6). It is precisely to avoid data dredging and post-hoc changes to categories and analysis in order to produce results that authors prefer that protocols should be published, and your paper is at risk of this. You say in the response to my comment that you did not publish a protocol to avoid data dredging. This cannot be true.

I am still not convinced that the tripartite classification of ISCED categories is the most appropriate. I accept that there is consideration precedent for this with the plethora of Mackenbach papers using it. However, this is very prone to differently sized groups meaning different things at different time points (as detailed in the Flanagan reference, which, incidently, is given in a very odd format and should be correct at proofing stage to make sure it is linked to the peer reviewed article).

Thank you for making the changes you have made to the paper, I think it is much improved as a result.

7. PLOS authors have the option to publish the peer review history of their article (what does this mean?). If published, this will include your full peer review and any attached files.

Reviewer #1: Yes: Gerry McCartney

---

## [Editor Report · Acceptance letter]

26 May 2020

PONE-D-19-34940R1 

Educational Inequalities in Mortality Amenable to Healthcare. A Comparison of European Healthcare Systems 

Dear Dr. Rydland:

I am pleased to inform you that your manuscript has been deemed suitable for publication in PLOS ONE. Congratulations! Your manuscript is now with our production department. 

With kind regards,

on behalf of

Prof. Dr. Brecht Devleesschauwer 

Academic Editor

PLOS ONE